# Regional Inversion of Soil Heavy Metal Cr Content in Agricultural Land Using Zhuhai-1 Hyperspectral Images

**DOI:** 10.3390/s23218756

**Published:** 2023-10-27

**Authors:** Hongxu Guo, Kai Yang, Fan Wu, Yu Chen, Jinxiang Shen

**Affiliations:** 1School of Architecture and Urban Planning, Guangdong University of Technology, Guangzhou 510090, China; guohx@gdut.edu.cn (H.G.); 2112210059@mail2.gdut.edu.cn (K.Y.); 2112210049@mail2.gdut.edu.cn (F.W.); 2International Research Center of Big Data for Sustainable Development Goals, Beijing 100094, China; 3Aerospace Information Research Institute, Chinese Academy of Sciences, Beijing 100094, China; 4School of Land and Space Information, Yunnan Land and Resources Vocational College, Kunming 652501, China; shenjx07@gmail.com

**Keywords:** soil heavy metal content, Zhuhai-1 hyperspectral images, direct standardization algorithm, optimal inversion model combination

## Abstract

With the development of hyperspectral imaging technology, the potential for utilizing hyperspectral images to accurately estimate heavy metal concentrations in regional soil has emerged. Currently, soil heavy metal inversion based on laboratory hyperspectral data has demonstrated a commendable level of accuracy. However, satellite images are susceptible to environmental factors such as atmospheric and soil background, presenting a significant challenge in the accurate estimation of soil heavy metal concentrations. In this study, typical chromium (Cr)-contaminated agricultural land in Shaoguan City, Guangdong Province, China, was taken as the study area. Soil sample collection, Cr content determination, laboratory spectral measurements, and hyperspectral satellite image collection were carried out simultaneously. The Zhuhai-1 hyperspectral satellite image spectra were corrected to match laboratory spectra using the direct standardization (DS) algorithm. Then, the corrected spectra were integrated into an optimal model based on laboratory spectral data and sample Cr content data for regional inversion of soil heavy metal Cr content in agricultural land. The results indicated that the combination of standard normal variate (SNV)+ uninformative variable elimination (UVE)+ support vector regression (SVR) model performed best with laboratory spectral data, achieving a high accuracy with an R^2^ of 0.97, RMSE of 5.87, MAE of 4.72, and RPD of 4.04. The DS algorithm effectively transformed satellite hyperspectral image data into spectra resembling laboratory measurements, mitigating the impact of environmental factors. Therefore, it can be applied for regional inversion of soil heavy metal content. Overall, the study area exhibited a low-risk level of Cr content in the soil, with the majority of Cr content values falling within the range of 36.21–76.23 mg/kg. Higher concentrations were primarily observed in the southeastern part of the study area. This study can provide useful exploration for the promotion and application of Zhuhai-1 image data in the regional inversion of soil heavy metals.

## 1. Introduction

Chromium (Cr) is a prevalent soil pollutant. Excessive Cr release into soil not only adversely impacts crop quality and human health but also jeopardizes the safety of the entire ecosystem [1,2]. Therefore, conducting a monitoring study on soil heavy metal Cr content is of great importance. Traditional field monitoring methods for soil heavy metal content are inefficient and expensive. These methods typically focus on monitoring specific points or soil profiles, making it challenging to accurately assess the spatial distribution of heavy metal content on a regional scale. However, with the development of hyperspectral imaging technology, it is possible to acquire large-scale and high-spectral-resolution remote-sensing images, which presents an opportunity for efficiently monitoring heavy metal pollution across extensive areas [3,4,5].

Currently, there are two main approaches for regional inversion of soil heavy metal content based on remote-sensing images: one that excludes the use of ground-measured laboratory spectral (GMLS) data and another that incorporates such data. The first approach involves the construction of models directly using soil sample heavy metal content data and remote sensing [6]. For instance, Guo et al. [7] utilized soil metal content data in conjunction with hyperspectral images from the GF-5 satellite to develop a random forest (RF) model for estimating the distribution of soil zinc (Zn) and nickel (Ni) concentrations. Yang Lingyu et al. [8] employed soil heavy metal content data and Hyperion images to create a partial least squares regression (PLSR) model, which they used to estimate the distribution of arsenic (As), lead (Pb), zinc (Zn), and cadmium (Cd). In a similar vein, Baihan et al. [9] used measured soil sample data and GF-5 hyperspectral satellite images to construct an extreme gradient boosting (XGBoost) inversion model, aided by genetic algorithm features, to assess the pollution status of heavy metal copper (Cu) in the Daxigou mining area of Shanxi Province, China. Liu et al. [10] employed hyperspectral images to accurately map the distribution of heavy metal contents (specifically As, Cd, and Hg) in the soil of Conghua District, Guangzhou, China. In addition to these models, support vector regression (SVR) [11,12] and artificial neural network (ANN) [8,13,14] are commonly used for this purpose. However, this approach is often associated with reduced reliability due to factors such as atmospheric and soil background interference during remote-sensing image acquisition. In light of these challenges, the second approach involves the integration of GMLS data. Many scholars have explored different spectral processing methods encompassing spectral pre-processing techniques such as Savitzky–Golay (SG) smoothing [15], SNV [16], multiplicative scatter correction (MSC) [17], first-order differential reflectance (FD) [18], and second-order differential reflectance (SD) [19]. Additionally, spectral selection methods such as principal component analysis (PCA) [20], competitive adaptive reweighted sampling (CARS) [21], UVE [22], and successive projections algorithm (SPA) [23] have been employed. These methods have significantly improved the accuracy of soil heavy metal inversion when referring to GMLS data in many studies. However, the variance in spectral resolution (i.e., the number of bands) between remote-sensing bands and GMLS data presents a challenge when applying these methods from GMLS data to remote-sensing images. Furthermore, the mixed-pixel effect complicates the direct application of models derived from ground spectra to remote-sensing images. Therefore, despite the successful models for soil heavy metal content estimation based on GMLS data, there are limited instances of their subsequent application to remote-sensing images.

In this study, Orbita hyperspectral (OHS) data from the Zhuhai-1 satellite, which has 32 spectral bands with a spatial resolution of 10 m, were used for regional soil heavy metal Cr content inversion combined with GMLS data in Shaoguan City, Guangdong Province, China. These rich spectral bands and high spatial resolution enabled us to integrate satellites with GMLS data for soil heavy metal inversion. The OHS data were calibrated using the direct standardization (DS) algorithm to ensure its alignment with the ground spectral data. Subsequently, by applying the optimal Cr inversion model derived from GMLS data and Cr content data, we achieve regional inversion of soil Cr content in agricultural land. The results can provide valuable technical support for the rapid and efficient monitoring of soil ecological conditions in agricultural land.

## 2. Materials and Methods

### 2.1. Study Area

The study area is located in a region highly impacted by Cr heavy metal pollution in Shaoguan City, Guangdong Province, China. Geographically, the center coordinate of the study area is 113.55° E longitude and 24.65° N latitude. Covering an area of 131.52 km^2^, the study area predominantly includes Zhangshi Town and its surrounding villages, characterized by a concentration of agricultural land parcels. The topography of the region primarily consists of mountains and basins, with higher elevations in the west and lower elevations in the east. Climatically, this area falls within the subtropical monsoon climate zone, marked by the prevalence of southwest and southeast monsoons during the summer and the influence of the northeast monsoon in winter. Since the 1950s, the establishment of heavy industries such as power plants, smelters, and steel factories has transformed this area, gradually making non-ferrous metal processing the dominant industry. Consequently, a substantial volume of industrial waste has been discharged into the soil, leading to serious heavy metal pollution in the agricultural land of this region [24,25]. In this study, an expansive area characterized by agricultural land and relatively flat terrain was deliberately selected. This choice aimed to minimize the potential impact of varying elevations and slopes on the experiment. Soil sampling was systematically conducted in areas with potential contamination, guided by the spatial distribution of the factories and the prevailing wind direction (Figure 1).

### 2.2. Workflow of Data Processing

This study consists of four main steps (Figure 2): (1) Data collection: The OHS images of the study area were acquired from the Orbit Zhuhai-1 remote sensing data service platform. We collected a total of 65 soil samples from various locations within the study area and then measured the concentration of the heavy metal Cr using graphite furnace atomic absorption spectrophotometry according to the GB/T 17141-1997 standard. These soil samples were then dispatched to the laboratory for analysis, where both the spectra and heavy metal content data of the samples were measured; (2) Model construction based on GMLS data: The spectral data and Cr content of 65 samples measured in the laboratory formed the foundation for model development. The dataset was partitioned into a training set and a validation set in a 7:3 ratio. The optimal combination of inversion models was determined through rigorous evaluation; (3) Hyperspectral image processing and calibration: Techniques, including geometry, radiation, and atmospheric calibration, as well as mixed-pixel decomposition methods, were employed to extract the bare land image from the OHS image. Additionally, the DS algorithm was applied to calibrate the OHS image with the GMLS data; (4) Regional mapping of Cr content: Building upon the optimal Cr inversion model derived from the previous step, we successfully achieved the regional soil heavy metal Cr content in the study area. The accuracy assessment was subsequently carried out through validation processes to verify the model’s performance.

### 2.3. Data Collection and Preprocessing

#### 2.3.1. Soil Sample Collection and Processing

In accordance with the spatial distribution characteristics of the local agricultural land, this study collected a total of 65 soil samples in July 2021. These 65 sampling points were strategically arranged, adhering to the principles of uniformity and representativeness. The spatial distribution of these sampling points is visually presented in Figure 1. Upon collection, the collected soil samples underwent a systematic process. They were first air-dried and subsequently ground in the laboratory before being sieved through a 20-mesh sieve. Following this preparation, each soil sample was divided into two equal parts. One part was dedicated to the measurement of Cr content, while the other was reserved for spectral measurements.

#### 2.3.2. Spectral Measurement and Preprocessing

The GMLS data were acquired using the ASD FieldSpec4 spectrometer (Analytical Spectral Devices Inc., Boulder, CO, USA). The spectral band range is 350–2500 nm, with a total of 2152 bands. For this study, we meticulously selected the 32 bands that corresponded to the Zhuhai-1 OHS data as the original soil spectral dataset. In the laboratory setting, the spectral measurements were conducted under controlled conditions. This involved employing a 1000 W halogen lamp as the light source. The observation angle between the light source and the vertical direction was set at 30°, with the distance between the light source and the sample consistently maintained at 30 cm. The probe was positioned vertically above the soil surface, maintaining a 5 cm distance from the soil’s surface. Each soil sample was measured in four directions, with rotations at 90° intervals each time. This process resulted in a total of 10 spectral curves being recorded for each direction. The actual spectral data utilized in this study was determined by averaging the 40 individual spectra obtained, ensuring the accuracy and reliability of the measurements.

During the process of acquiring soil spectra, various factors such as acquisition instruments, soil moisture, and external environment conditions can introduce interference into the spectral data. This interference can lead to a spectral curve containing unwanted noise, which will seriously affect the inversion accuracy. To mitigate this challenge, spectral preprocessing techniques were employed to eliminate the noise and enhance the spectral characteristics specific to soil heavy metal Cr. In this study, the first step involves the application of the Savitzky–Golay (SG) smoothing filter to eliminate the noise present in the original spectral data. Subsequently, six distinct spectral transformation methods were employed: SNV, MSC, FD, SD, discrete wavelet transform (DWT), and reciprocal logarithm (RL). These transformation methods served to highlight subtle differences within spectral curves and enhance the spectral features associated with soil heavy metal Cr [26,27]. Finally, the Pearson correlation coefficient (PCC) was utilized. PCC quantifies the strength of the linear correlation between variables X (the spectra) and Y (the soil heavy metal content). Its values range from −1 to 1, with larger absolute values indicating stronger correlations [28,29]. The method that demonstrates the highest correlation with the soil heavy metal content is selected, and feature bands are subsequently extracted from the transformed spectra using this chosen method. This approach ensures the effective enhancement of relevant spectral characteristics while minimizing noise, ultimately contributing to the accuracy of the study’s findings.

#### 2.3.3. Image Data Acquisition and Preprocessing of Zhuhai-1

In this study, we employed Zhuhai-1 OHS data [30], which can be obtained at https://www.obtdata.com/#/index (accessed on 15 February 2023). The data comprises 32 spectral bands, characterized by a band interval of either 14 nm or 16 nm. The spectral resolution ranges from 400 nm to 1000 nm, and the spatial resolution is 10 m. Compared to widely used hyperspectral satellite data such as Hyperion, Modis, and HJ-1, the OHS image has a relatively higher spatial resolution (10 m). This increased spatial resolution plays a crucial role in reducing the influence of mixed-pixel effects on the results of heavy metal inversion from soil. Furthermore, it is available free of charge. The detailed parameters are shown in Table 1.

The OHS data underwent essential preprocessing steps, including geometric correction, radiometric calibration, and atmospheric correction. Geometric correction ensured that image errors were kept within half a pixel. Radiometric scaling converted a pixel’s digital number (DN) value into apparent reflectance data. Since the OHS images do not include atmospheric correction parameters, a swift atmospheric correction method was applied to mitigate the influence of factors such as water vapor, carbon dioxide, various atmospheric molecules, and aerosols. This correction process effectively retrieves surface reflectance data, enhancing the quality and reliability of the dataset for subsequent analysis.

In addition, the spatial resolution of the OHS images is 10 m, resulting in mixed pixels that encompass various ground features such as buildings and vegetation. This significantly impacts the accuracy of soil spectral information. To effectively tackle this challenge, a two-step process was undertaken. Firstly, through a combination of visual interpretation and on-site verification, distinct end members were identified on the images, representing pure soil, water, vegetation, clouds, and building land pixels. Subsequently, a fully constrained least squares mixed-pixel decomposition method was applied to these pixels. This approach yielded abundance maps for each of the land cover types [31,32]. The abundance within each pixel represents the proportion of each land cover type. The resulting soil abundance map is shown in Figure 3a, with pixels exhibiting soil abundance values surpassing 0.6 being earmarked as agricultural land for the regional inversion of soil heavy metals [33], as illustrated in Figure 3b.

### 2.4. Image Spectral Calibration Based on Direct Correction (DS) Algorithm

To mitigate the impact of uncertain environmental factors, the DS algorithm was employed to calibrate the OHS image spectrally. The DS algorithm is a common, simple, and effective method for removing environmental factors from spectral data. It calculates a transfer matrix based on the relationship between satellite spectra and laboratory spectra, allowing for the correction of satellite spectra to mitigate the influence of environmental factors like moisture, particle size, and temperature. Several studies have successfully employed the DS algorithm to correct satellite spectra, yielding favorable correction results [34,35,36]. The transfer matrix was determined by ArcGIS 10.3 software to extract the corresponding spectral data (X_OHS_) of the sampling point on the OHS bare land images and corresponding GMLS data (X_Lab_). This matrix quantifies the deviation between OHS image spectral and GMLS data.
(1)$$X_{\mathrm{Lab}}=X_{\mathrm{OHS}}B+E\;$$
where B represents the transfer matrix between X_Lab_ and X_OHS_, and E is the residual matrix. If the parameters B and E of the DS algorithm are obtained, the entire OHS image spectral can be calibrated to spectra similar to the GMLS data through B and E.
(2)$$X_{OHS}^{’}=X_{OHS}B+E$$
where X’_OHS_ is the spectra of the OHS image spectral calibrated by the DS algorithm.

In addition, it is important to note that the representativeness and number of transfer samples can influence the performance of the DS algorithm, subsequently affecting the accuracy of the inversion model. To address this, the Kennard-Stone (KS) algorithm was employed to select a series of transfer samples from the pixels corresponding to the 26 bare soil sampling points. The number of transfer samples, which yields the best performance of the DS algorithm, was determined based on the inversion accuracy, assessed by the determination coefficient R^2^ [37].

### 2.5. Band Selection and Modeling

Previous research has underscored the importance of band selection and modeling methods in achieving precise hyperspectral inversion estimates of soil heavy metals. In this study, the KS algorithm was employed to partition the dataset, with 70% of the samples allocated to the training set for model development, and the remaining 30% earmarked as the verification set for assessing the model accuracy. In order to estimate the content of heavy metal Cr with high precision, four feature band selection algorithms, namely PCA [38], CARS [39], UVE [40], and SPA [41], were systematically combined with four modeling algorithms, including SVR [42], BPNN [43,44], PLSR [45,46] and RF [47,48], in various combinations (Figure 4). By comparing the accuracy of these diverse combinations, the optimal inversion model for heavy metal Cr was pinpointed.

### 2.6. Model Accuracy Evaluation

In this study, four metrics including coefficient of determination (R^2^), root mean square error (RMSE), mean absolute error (MAE), and relative analysis error (RPD) [49] were used for evaluating the performance of the inversion model. The R^2^ reflects the fitting degree of the model, with values ranging from 0 to 1. A value closer to 1 indicates a higher degree of fit. RMSE represents the deviation between the predicted value and the true value. A smaller RMSE value signifies lower prediction error and higher model accuracy. MAE is the average value of the absolute value of all individual observations deviating from the arithmetic mean. A smaller MAE value indicates a superior predictive capacity of the model. The reliability of the model is assessed based on RPD. When RPD ≥ 2, the model is considered reliable. When 2 > RPD ≥ 1.5, the model is relatively reliable. If RPD < 1.5, the model is deemed unreliable [50,51].

## 3. Results

### 3.1. Descriptive Statistics of Cr Content

Descriptive statistics of the Cr content (mg/kg) measured from the in situ samples are shown in Table 2 below. The Cr content ranged from 8 to 139 mg/kg, with an average concentration of 47.28 mg/kg. When examining the spatial distribution, the coefficient of variation falls within the range of 0.5 to 0.75, indicating a moderate level of variability. This implies that the distribution of Cr in the soil is not uniform, and there exists significant spatial variability in its occurrence.

### 3.2. Optimal Inversion Model Combination

#### 3.2.1. Optimal Spectral Transformation Method

PCC analysis was conducted on the Cr content and the original spectral data, as well as the spectral data after six transformations of SNV, MSC, FD, SD, DWT, and RL, using MATLAB. Figure 5 shows the results of the correlation analysis, which demonstrates that there is some improvement in the correlation after several spectral transformation processes. This indicates that the spectral transformation can effectively mitigate the impact of noise originating from factors such as instrument variations and external conditions. Among these spectral transformation methods, as shown in Table 3, SNV exhibited the highest maximum correlation coefficient value, achieving −0.804 at 922 nm, indicating an overall superior performance. Therefore, for this study, SNV transformation data were ultimately selected as the foundation dataset for feature band selection.

#### 3.2.2. Optimal Inversion Model Combination

After applying the SNV transformation, four feature band selection methods—PCA, CARS, UVE, and SPA—were employed to extract the sensitive spectral response band associated with the Cr element. These methods yielded 6, 11, 7, and 10 feature bands, respectively, with most of them concentrated in the 866–940 nm wavelength range. Subsequently, the feature bands selection by these algorithms were utilized to estimate the Cr content using the SVR, BPNN, PLSR, and RF models respectively. The inversion results are shown in Table 4. Following a comparative analysis of model accuracy, the optimal model combination for Cr content inversion was determined to be SNV + UVE + SVR. The training model accuracy R^2^, MAE, and RMSE were 0.89, 7.55, and 11.60, respectively. Meanwhile, the verification model exhibited the following accuracy metrics: R2 of 0.97, RMSE of 5.87, MAE of 4.72, and RPD of 4.04. Notably, R^2^ = 0.97 > 0.66 and RPD = 4.04 > 2, indicating that the model combination has strong stability and reliability, and can be used for regional inversion of soil heavy metal Cr content.

### 3.3. DS Transfer Set Size Setting and Calibration

The performance of the DS algorithm is closely related to the number of samples in the transfer set. The determination coefficient R^2^ in the inversion accuracy of Cr content is used to determine the optimal DS transfer set size. The R^2^ values associated with different transfer set sizes are shown in Figure 6d. It can be seen from Figure 6d that the R^2^ value reaches its pinnacle when the transfer set size is 21. Consequently, this study has chosen to utilize 21 samples for calculating the transfer matrix.

Figure 6 presents a notable observation, where the spectral features of the 21 OHS image samples, calibrated by the DS algorithm (Figure 6b), closely resemble the corresponding GMLS data (Figure 6a). The results underscore the effectiveness of the DS algorithm in mitigating the inevitable environmental influences such as soil moisture content, soil particle size, and meteorological factors. The obtained transfer matrix can be applied to calibrate the entire OHS image spectra.

### 3.4. Inversion of Spatial Distribution of Heavy Metal Cr

Based on the optimal model combination (SNV + CARS + SVR) of the GMLS data and the calibrated entire OHS image spectral data, heavy metal Cr content inversion was carried out in the bare land area of the study area. The inversion result was categorized into five levels by using the natural discontinuity method, as shown in Figure 7. The result illustrates that the majority of Cr content falls within the range of 36.21–76.23 mg/kg. Higher concentrations of Cr elements were observed in the southeastern and southwestern part of the study area, ranging from 110.53 to 149.41 mg/kg, while the lower Cr content is primarily distributed in the northern region. This distribution may be closely related to the location of heavy industry enterprises in the area, the prevailing wind direction, and the flow direction of rivers. Enterprises are mainly situated in the northeastern part of the study area, with industrial emissions released through elevated chimneys. These emissions are influenced by the prevailing wind direction (northeast monsoon), and particles containing heavy metal elements are transferred by wind and rainfall, resulting in higher Cr element content in the southwestern part of the study area. Furthermore, enterprises are mainly situated in the upper reaches of the river, and industrial waste is transported by flowing water and adsorbed by sediment, leading to higher Cr element content in the lower reaches of the river, which is distributed in the southeast of the study area.

Twenty sets of inversion values and measured values were randomly selected to evaluate the accuracy of the regional inversion results for Cr content. The coefficient of determination (R^2^) and root mean square error (RMSE) were utilized for this analysis. The accuracy verification results (Figure 8) revealed an R^2^ value of 0.63 and an RMSE value of 20.20. These results indicate a reasonable level of reliability in the regional inversion results for soil heavy metal content in the study area [6].

According to the “Soil Environmental Quality Agricultural Land Soil Pollution Risk Control Standards (Trial)” (GB 15618-2018) of the People’s Republic of China, the pH values of the sampling points were examined. The majority of pH values in the study area were found to be below 6.5. Consequently, the corresponding risk screening value for Cr content was selected based on two pH ranges: PH ≤ 5.5 and 5.5 < PH ≤ 6.5. The risk screening value for both ranges is 250 mg/kg. It is worth noting that the highest inversion value obtained for Cr content in this study is 149.41 mg/kg, which falls below the risk screening value for paddy fields and other agricultural uses of land.

## 4. Discussion

### 4.1. The Impact of Mixed Pixels

When employing satellite imagery for the inversion of soil heavy metal content, the predominant challenge is effectively addressing the issue of mixed pixels. Field soil samples are typically collected from areas of bare soil, which are then combined with satellite image data from the same time period. As spatial resolution decreases, the likelihood of other interfering surface features coexisting within a single pixel increases. There must be a sufficient number of bare soil pixel samples available to meet the modeling requirements. This issue becomes particularly pronounced in regions with abundant water and favorable thermal conditions, coupled with dense vegetation cover. With a spatial resolution of 30 m to 1 km, traditional hyperspectral satellite data such as Hyperion, Modis, and HJ-1 are difficult to ensure that all image pixels corresponding to the sampling points represent pure bare soil, as they are frequently affected by surrounding factors like vegetation, built-up areas, and water bodies. The Zhuhai-1 OHS image has a relatively higher spatial resolution (10 m) which plays a crucial role in reducing the influence of mixed-pixel effects on the results of heavy metal inversion from soil. Nevertheless, even though our sampling locations were positioned within areas of bare soil during collection, a slight presence of vegetation cover persisted within the corresponding OHS data pixels captured in some of the samples. We anticipate a substantial improvement in this regard with the enhancement of spatial resolution in hyperspectral imagery.

### 4.2. Challenges of Model Generalization

In this study, many data processing methods, such as DS conversion of satellite images and combined transformation of spectral data were utilized for developing an optimal model. While these methods have improved the accuracy of the model, they also present challenges for the generalization of the model to other research cases. Establishing a robust inversion model necessitates a larger sample size to support the data. In this experiment, we observed a significant correlation between Cr content and the wavelength range of 866–940 nm in the near-infrared spectra. Interestingly, this band corresponds to the feature band position of typical substances found in soil, such as organic carbon, nitrogen, and clay minerals. Whether the spectral response of Cr in soil is attributed to the difference in the material composition of the adsorbed soil, and to what extent this difference affects the model’s generalizability, requires further experiments for exploration. Additionally, we did not incorporate factors like soil physical and chemical properties, such as pH value, particle size, and moisture content, into the modeling process. Nevertheless, these factors may exert a potential influence on the inversion model and warrant further comprehensive investigation. Future research endeavors could contemplate the inclusion of these soil physical and chemical properties in the modeling process to enhance the model’s applicability and transferability.

## 5. Conclusions

This study conducted satellite hyperspectral image acquisition and indoor soil sample collection in typical Cr-polluted areas in Shaoguan, China. The optimal inversion model combination was obtained by applying spectral preprocessing, feature band selection, and model construction techniques to the spectra and Cr content data of indoor soil samples. The result shows that the optimal model combination for Cr content inversion is SNV + UVE + SVR. The model exhibits high accuracy, with R^2^, RMSE, MAE, and RPD values of 0.97, 5.87, 4.72, and 4.04, respectively, indicating its stability and reliability. Furthermore, we employed the DS algorithm to eliminate the influence of environmental factors on the image spectra, and in conjunction with the OHS images, we successfully inverted the regional Cr content of the study area. The Cr content ranged from 36.21 to 149.41 mg/kg, and the accuracy of R^2^ and RMSE was 0.63 and 20.20, respectively, indicating a high level of reliability in the inversion results.

This study demonstrated the feasibility of using hyperspectral images to monitor soil heavy metal content on a regional scale by establishing an inversion model based on laboratory data. The workflow and methodology employed in this study provide valuable scientific support for such work.

## Figures and Tables

**Figure 1 sensors-23-08756-f001:**
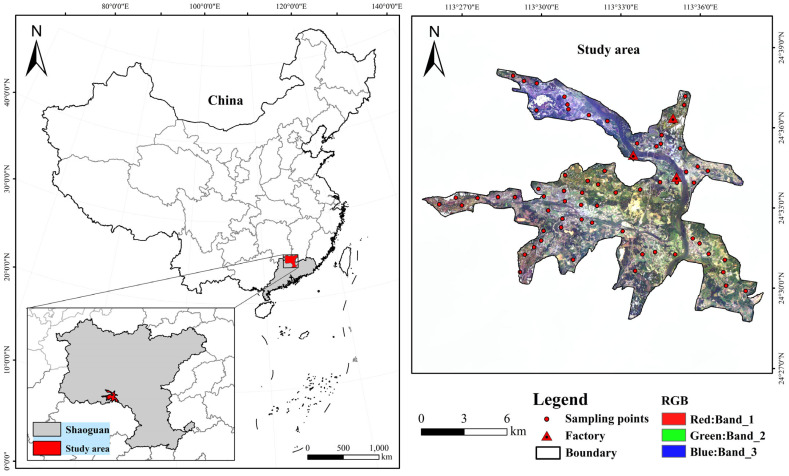
Location of the study area and sampling distributions (the figure on the right displays an OHS band composite image from the Zhuhai-1 satellite with Bands 1, 2, and 3 assigned to the RGB channels, respectively).

**Figure 2 sensors-23-08756-f002:**
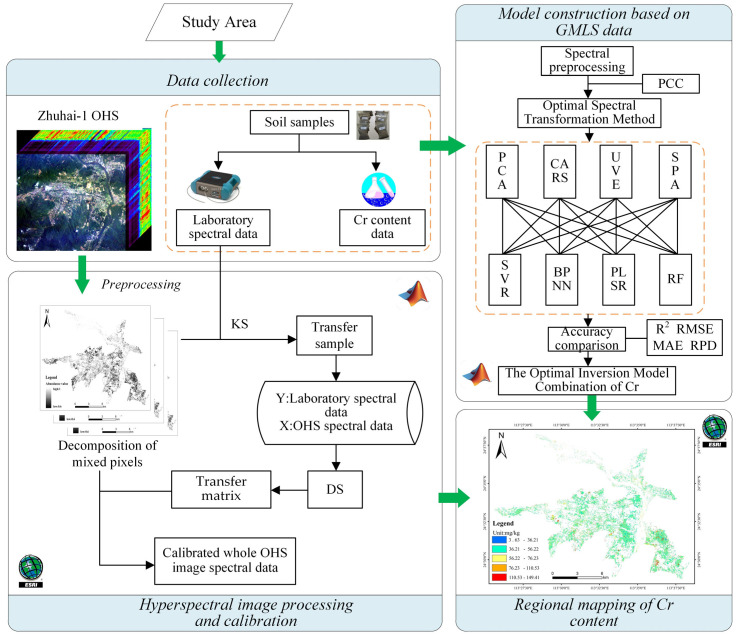
The flowchart of this study.

**Figure 3 sensors-23-08756-f003:**
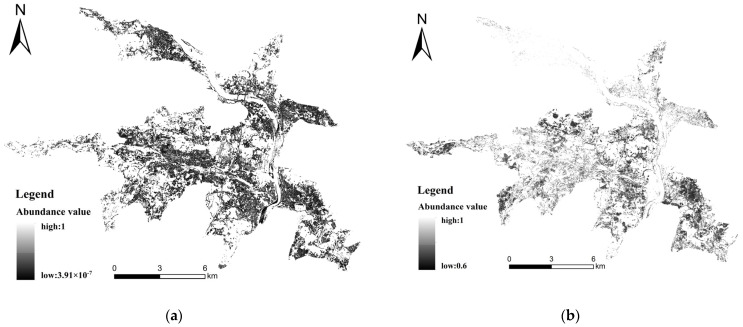
Bare land abundance map before extraction (**a**) and abundance map after extraction (**b**).

**Figure 4 sensors-23-08756-f004:**
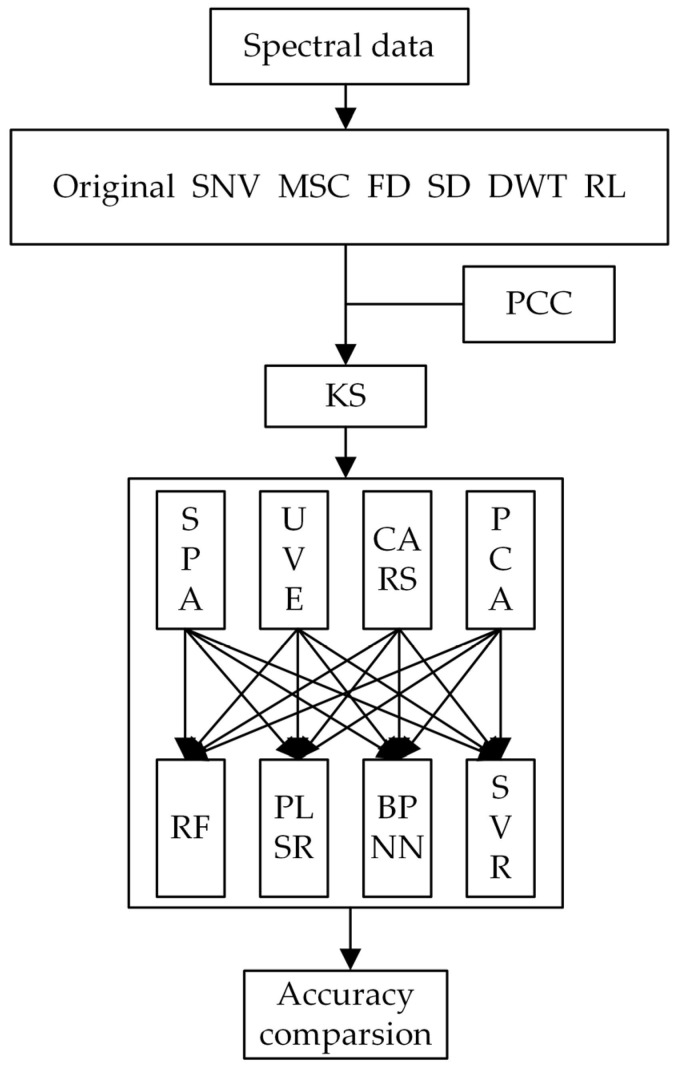
Inversion model combination.

**Figure 5 sensors-23-08756-f005:**
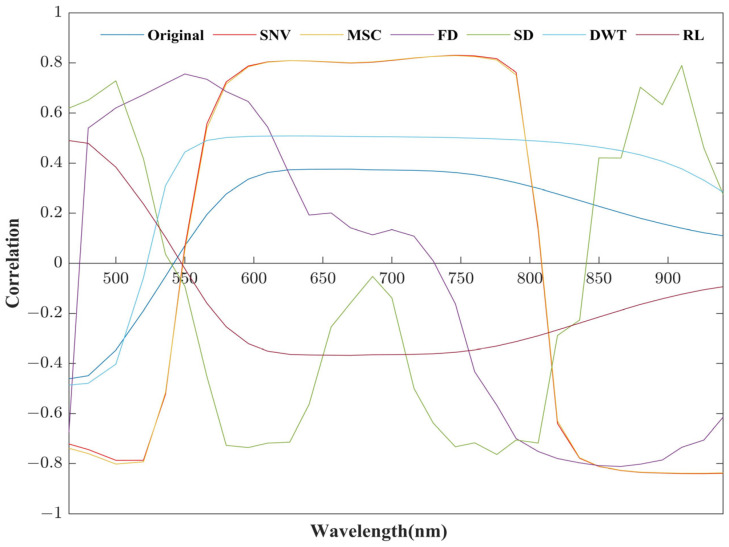
Correlation curve between Cr content and different transformations of spectral data.

**Figure 6 sensors-23-08756-f006:**
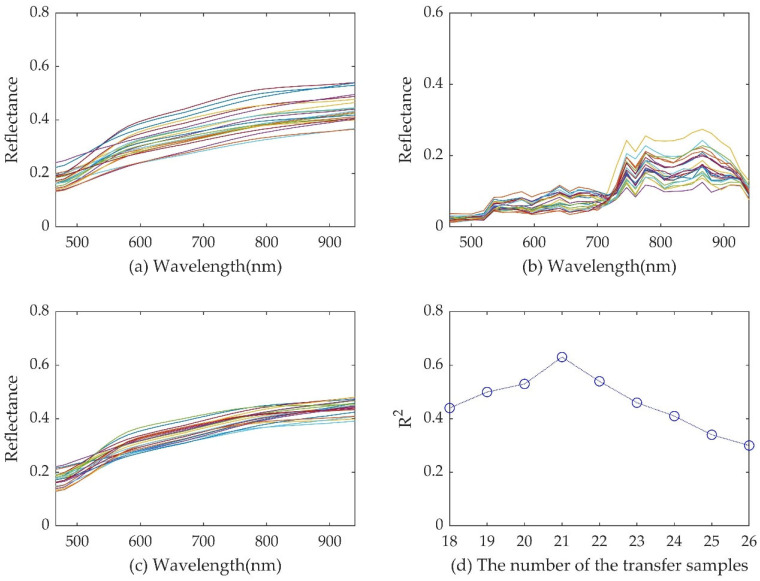
GMLS data curve (**a**), images spectral curve (**b**), corrected image spectra (**c**), and modeling accuracy corresponding to different numbers of transfer sets (**d**).

**Figure 7 sensors-23-08756-f007:**
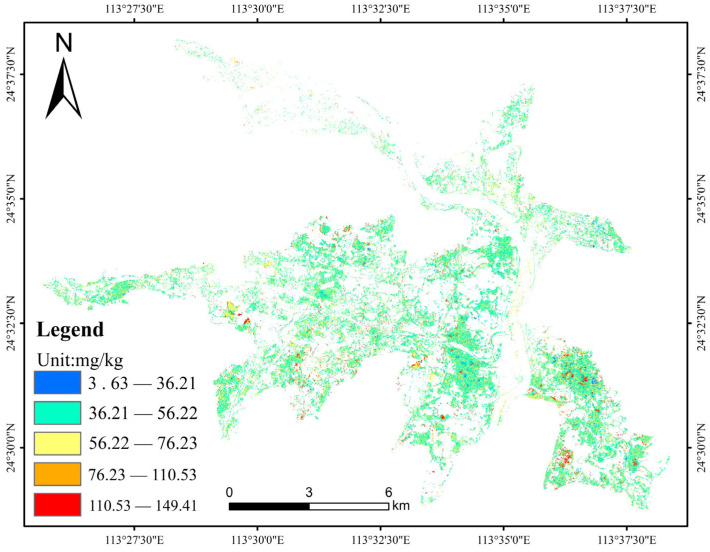
Spatial distribution of heavy metal Cr content.

**Figure 8 sensors-23-08756-f008:**
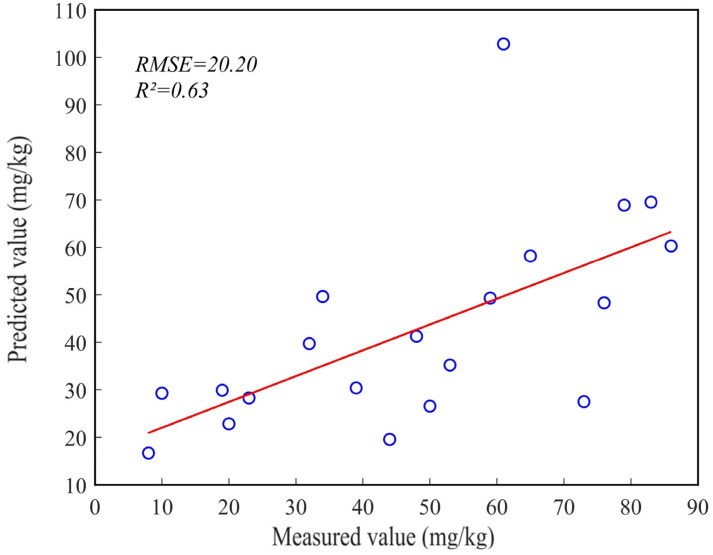
Accuracy verification result.

**Table 1 sensors-23-08756-t001:** Zhuhai-1 satellite parameters.

Parameter		Value
Center longitude and latitude	Lon	113.500775
	Lat	24.495786
Sensor		CMOS
Imaging time		27 July 2021
Solar elevation angle		62.59
Side swing angle		3.40
Ground resolution		10 m
Track height		520 km
Cloud cover		5%
Data level		L1B

**Table 2 sensors-23-08756-t002:** Descriptive statistics of Cr content in the study area.

Metal	Number	Mean (mg/kg)	Standard Deviation (mg/kg)	CV (%)	Minimum (mg/kg)	Maximum (mg/kg)	Skewness	Kurtosis	Topsoil of Guangdong Province	Topsoil of China
Cr	65	47.28	25.18	53	8	139	0.75	1.08	54.91	54

**Table 3 sensors-23-08756-t003:** Statistical analysis of correlation coefficient.

Spectral Transformation	Band (nm)	Maximum Correlation Coefficient
Original	468	−0.549
SNV	922	−0.804
MSC	922	−0.802
FD	840	−0.792
SD	474	0.688
DWT	460	−0.516
RL	468	0.573

**Table 4 sensors-23-08756-t004:** Results of different band selections and different inversion models.

Preprocessing	Band Selection	Modeling Method	Training Set			Validation Set			
			R^2^	RMSE	MAE	R^2^	RMSE	MAE	RPD
SNV	PCA	SVR	0.90	10.86	7.74	0.91	10.43	8.58	2.27
		BPNN	0.89	12.16	9.44	0.90	11.68	9.12	2.03
		PLSR	0.84	13.77	10.99	0.94	10.70	8.37	2.21
		RF	0.85	13.71	9.23	0.94	8.66	6.54	2.74
	CARS	SVR	0.81	14.96	11.63	0.92	11.07	8.41	2.14
		BPNN	0.86	12.86	10.07	0.91	10.89	8.30	2.18
		PLSR	0.82	14.65	11.96	0.85	12.01	9.13	1.85
		RF	0.85	13.74	9.56	0.89	11.88	9.79	1.99
	UVE	SVR	0.89	11.60	7.55	0.97	5.87	4.72	4.04
		BPNN	0.70	19.17	13.44	0.83	14.32	12.74	1.66
		PLSR	0.83	14.21	11.68	0.86	11.58	9.16	1.92
		RF	0.84	13.75	9.72	0.95	7.90	6.06	3.00
	SPA	SVR	0.78	15.83	11.97	0.92	9.36	7.22	2.36
		BPNN	0.89	11.55	9.17	0.93	8.60	6.73	2.76
		PLSR	0.85	13.30	11.38	0.85	12.39	8.96	1.91
		RF	0.85	13.37	9.20	0.95	8.04	6.36	2.95

## Data Availability

The data that support the finding of this study are available from the corresponding author upon reasonable request. The data are not publicly available due to restrictions eg privacy.

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
