# Peer review of "Regional Inversion of Soil Heavy Metal Cr Content in Agricultural Land Using Zhuhai-1 Hyperspectral Images"

_sensors, 2023, doi:10.3390/s23218756_

Round 1
Reviewer 1 Report
Comments and Suggestions for Authors
1 part 3.1 how many samples were used to detect Cr content?
2 please develop quality of Figure 5,
3 please explain abnormal value of Figure 8
Comments on the Quality of English Language
need minor revise
Author Response
Thank you very much for the suggestions and comments, which have great contributed to the improvement of the manuscript. We have carefully revised the manuscript and addressed each of the comments. Please see the attachment.

Reviewer 2 Report
Comments and Suggestions for Authors
Please see the attachment peer-review-32180066.v1.docx

Author Response

(The authors gave the same response as above.)

Reviewer 3 Report
Comments and Suggestions for Authors
Abstract: The authors aims and scope of the paper are not clearly defined within the Abstract. Was the aim to determine soil Cr content with satellite images? Or the aim is to model the bands of the hyperspectral satellite to bands of the laboratory instrument to assure a more accurate determination of Cr by the satellite? This must be clearly presented.
If the scope of the paper is to “synchronize” the two spectral sensors (satellite and lab) why not try to do that with a first sampling campaign and then do another sampling campaign to validate the results.
To my opinion, 36 samples cannot be considered as a sufficient number of samples for both a calibration and a validation.
The authors should explain and justify (in the paper) the reason for receiving 36 samples. Also how did the authors plan the sampling campaign (selection of areas, representative samples etc.).
L22: Is the word inversion here, used as "determination"? Justify the reason you have selected the word “inversion”.
L22: Why is the word regional highlighted here? This method presents no applicability – scalability to other regions?
L23: What the abbreviations stand for?
L30-31: If the model was applied in another area - not the study site - with an acceptable accuracy, then this assumption would be valid. Change or remove this sentence. Your findings do not support this hypothesis.
L96: How are these two factors affect the sampling. Elaborate and justify.
Does it mean that you also received soil samples from vegetated areas? Does that mean that the Satellite was depicting vegetated areas? How can satellite images depicting vegetation be matched with soil data? The authors should explain thoroughly.
L107: When was the date of sampling and when of image acquisition?
Also, you removed the vegetation reflectance to keep the bare soil reflectance from a mixed pixel? Is that method accurate?
And another question is why you did not receive only samples from bare soil and selected vegetated areas as well?
L143: Is this only an enhancement for Cr? What if other trace elements are also enhanced? How do these 6 methods can only discriminate the Cr? Explain thoroughly.
L165: Provide examples of other hyperspectral satellites that could also be used to show the scalability of the proposed method. Could your method be reproduced? Your contribution to the scientific community is not evident and should be enforced within the paper.
Comments on the Quality of English LanguageModerate English language editing is required.
Author Response

(The authors gave the same response as above.)

Round 2
Reviewer 3 Report
Comments and Suggestions for Authors
A lot of concerns regarding the followed methodology have been clarified and texts to justify many missing points have been provided.
However, there is a certain remark considering the structure of the document. The Discussion section is very limited and needs significant additions. What is lacking from the Discussion, is the reference to studies and findings of other researchers, and a comparison of the current study to previous relative research.
Therefore, I suggest the authors to enrich this part.
In view of this, I recommend a minor revision to allow the authors to make the appropriate additions.
Comments on the Quality of English LanguageModerate editing of English language required
Author Response
Thank you very much for your suggestion. We further enriched the content of the discussion part from the "impact of mixed pixels"and "Challenges of model generalization". Line 431-470.